# The association of sun exposure, ultraviolet radiation effects and other risk factors for pterygium (the SURE RISK for pterygium study) in geographically diverse adult (≥40 years) rural populations of India -3rd report of the ICMR-EYE SEE study group

Radhika Tandon[1]ᵒ*, Praveen Vashist[1]ᵒ, Noopur Gupta[1], Vivek Gupta[1], Saumya Yadav[1], Dipali Deka[2], Sachchidanand Singh[3], K. Vishwanath[4], G. V. S. Murthy[5,6]

1 Dr. Rajendra Prasad Centre for Ophthalmic Sciences, AIIMS, New Delhi, India, 2 Regional Institute of Ophthalmology, Guwahati, India, 3 CSIR-National Physical Laboratory, New Delhi, India, 4 Pushpagiri Vitreo Retina Institute, Secunderabad, Telangana, India, 5 Indian Institute of Public Health, Hyderabad, India, 6 Clinical Research Department, Public Health Eye Care & Disability, London School of Hygiene & Tropical Medicine, London, United Kingdom

ᵒ These authors contributed equally to this work.
* radhika_tan@yahoo.com

## Abstract

### Purpose

To determine the prevalence and risk factors for pterygium in geographically diverse regions of India.

### Methods

A population-based, cross-sectional multicentric study was conducted in adults aged ≥40 years in plains, hilly and coastal regions of India. All participants underwent a detailed questionnaire-based assessment for sun exposure, usage of sun protective measures, exposure to indoor smoke, and smoking. Detailed ocular and systemic examinations were performed. Pterygium was diagnosed and graded clinically by slit-lamp examination. Association of pterygium with sociodemographic, ophthalmological, and systemic parameters was assessed. Physical environmental parameters for the study period were estimated.

### Results

Of the 12,021 eligible subjects, 9735 (81% response rate) participated in the study. The prevalence of pterygium in any eye was 13.2% (95% CI: 12.5%-13.9%), and bilateral pterygium was 6.7% (95% CI: 6.2–7.2). The prevalence increased with age (<0.001) irrespective of sex and was highest in those aged 60–69 years (15.8%). The prevalence was highest in coastal (20.3%), followed by plains (11.2%) and hilly regions (9.1%). On multi-logistic

**Data Availability Statement:** All data files are available from the Figshare repository: Tandon, Radhika; Vashist, Praveen; Gupta, Noopur; Gupta, Vivek; Yadav, Saumya; Deka, Dipali; et al. (2022): ICMR_Pterygium_Study_Data.csv. figshare. Dataset. https://doi.org/10.6084/m9.figshare.19626003.v1.

**Funding:** Indian Council of Medical Research (ICMR), India provided funding for this research project (Grant No 68/4/2009-NCD-1). The funders had no role in study design, data collection and analysis, decision to publish, or preparation of the manuscript.

**Competing interests:** The authors have declared that no competing interests exist.

regression, pterygium was positively associated with coastal location (P<0.001), illiteracy (P = 0.037), increasing lifetime sun exposure (P<0.001), and negatively associated with BMI $\geq$25 kg/m2 (P = 0.009).

## Conclusion

Pterygium prevalence is high in the rural Indian population. The association of pterygium with several potentially modifiable risk factors reflects its multifactorial etiology and provides targets for preventive measures.

## Introduction

Pterygium is a common ocular disorder characterized by a 'triangular encroachment of bulbar conjunctival tissue onto the cornea'. The earliest mention of pterygium can be dated back to the ancient texts by Susruta Samhita and Hippocrates [1, 2]. Since then, extensive epidemiological studies have been undertaken in an attempt to understand the etiopathogenesis of pterygium and to identify its risk factors [3–8]. Pterygium was included in the World Health Organization's priority eye conditions due to their impact on vision, quality of life and burden on healthcare systems [9]. Hence, appropriate strategies need to be implemented along with generation of evidence-based epidemiological data for detailed planning, monitoring and evaluation of interventions for this important public health problem.

Although several risk factors, including geographic location and climate, have been studied, excessive and prolonged sunlight exposure, particularly ultraviolet (UV) radiation, remains the most important [10, 11]. Apart from the geophysical elements, several socioeconomic, lifestyle, and systemic factors have been hypothesized to play either a direct or indirect role in the pathogenesis of pterygium. India is a large country with diverse geographical and climatic conditions ranging from tropical in the south to temperate and alpine in the Himalayan north. Variations in socioeconomic status and lifestyle patterns exist across locations. Hence, the current study was planned to determine and compare the prevalence of pterygium in multicentric geographically diverse locations of India including populations from plains, hilly and coastal areas and explore the interplay of risk factors in its pathogenesis.

## Materials and methods

A multicentric, population-based, cross-sectional study was conducted at three geographically diverse locations in the rural Indian population between 2010 and 2016 in individuals aged $\geq$40 years [12, 13]. The three study sites were diligently chosen to represent plains, hilly and coastal areas of the country. Gurugram district, Haryana State of National Capital Region (NCR) Delhi was chosen as representative for northern plains (henceforth referred to as Delhi NCR). The study in north-eastern hills was conducted at Kamrup district located adjacent to Guwahati, the capital of Assam (henceforth referred to as Guwahati). Prakasam district on the eastern coast line of Andhra Pradesh State was chosen to represent the southern coastal region. The study adhered to the Declaration of Helsinki. The study was approved by Institute Ethics Committee, All India Institute of Medical Sciences, New Delhi, India (P-16/04.08.2009); Indian Institute of Public Health, Hyderabad, India (33/2011–08–08); and Regional Institute of Ophthalmology, Guwahati, India (MC/190/2007/ 1098–23.02.2010). Written informed consent was obtained from all participants before enrolment in the study.

## Participant recruitment and screening

The selection and recruitment of participants along with protocol for detailed ophthalmological and systemic examination has been previously described in detail [12, 13]. To summarize, house visits were conducted by a trained health worker in the randomly selected clusters, and house members were interviewed using a structured questionnaire schedule. Eligible participants (≥40 years) were invited to come for a detailed ophthalmic examination at a local indoor clinic set up at the study site.

All ophthalmic examinations were carried out by an ophthalmologist. A portable slit-lamp was used for anterior segment examination. Pterygium was identified as a triangular fleshy mass extending from bulbar conjunctiva and encroaching onto the cornea. Pterygium was graded clinically depending on the extent of corneal involvement by the head of pterygium; Grade I—between limbus and a point midway between limbus and pupillary margin, Grade II —between a point midway between limbus and pupillary margin and pupillary margin (nasal pupillary margin in the case of nasal pterygium and temporal margin in the case of temporal pterygium) and Grade III—crossing pupillary margin [14]. Pterygium was labelled as double-headed in the cases with both nasal and temporal involvement. The systemic examination included measurement of height, weight, random blood sugar and blood pressure [12, 13].

## Conjunctival Ultraviolet Autofluorescence (CUVAF) imaging system

A custom-built camera system was used for recording CUVAF images as per previously described specifications [15]. A height adjustable table was equipped with a subject headrest, camera positioning assembly, digital single-lens reflex camera, macro lens, and filtered electronic flash. Three consecutive pictures of nasal and temporal region of each eye were captured, and the image with best clarity was selected for further assessment. Images were saved in RGB format at the D100 settings of JPEG (1:4 compression). The autofluorescence area in $mm^2$ on UV autofluorescence photographs was calculated using ImageJ software by two experienced graders masked by clinical findings. In eyes where multiple discrete areas of AF were present, each area was calculated separately, and the total area was represented as summation of these.

## Sun exposure and climatic parameters

The lifetime effective sun exposure was calculated for every individual using the formula based on the Melbourne visual impairment project model [16]. Satellite-based data was used for the long-term UVA (315–400 nm), UVB (280–315 nm), and aerosol optical depth (AOD) values at each three locations. In addition, meteorological data for humidity, precipitation, temperature, wind speed, and air pollutants were also obtained for the three locations [12, 13].

## Statistical analysis

Double entry of all data was done in a Microsoft Access™ database to avoid transcription errors. Data was analyzed using Stata 13 (Stata- Corp, College Station, TX). Participants with incomplete information on sun exposure or ocular examination were excluded. The compiled data on Microsoft Excel spreadsheet (Microsoft, Redmond, WA) was analysed using Statistical Package for the Social Sciences (SPSS) software (version 20; IBM Corp., Armonk, NY). All study participants were categorized into quintiles based on the lifetime effective sun exposure. Pearson chi-square test, t-test and Kruskal-Wallis tests were used for data that was categorical, continuous, and non-parametric continuous respectively. Risk factor comparisons were performed within-site and for combined data. P value < 0.05 was considered statistically

significant and 95% confidence intervals (CI) were calculated. Continuous variables were assessed and summarized using mean (standard deviation). Categorical variables were assessed using chi-squared test. Non-normal continuous variables were assessed using the Wilcoxon–Mann–Whitney rank-sum test.

## Results

Of the 12,021 eligible subjects enumerated at the three study sites, 9735 individuals (81% response rate) ≥40 years of age underwent detailed risk factor analysis and clinical assessment for pterygium (Delhi NCR- 3,595; Guwahati- 3,231; Prakasam- 2,909) (Fig 1). Socio-demographic and baseline clinical characteristics of participants of the ICMR- EYE SEE study have been previously reported in detail [12, 13]. The climatic parameters during the conduct of the study at each of the three study locations have also been highlighted [12, 13].

The overall prevalence of pterygium in either eye was 13.2% (95% CI: 12.5%-13.9%; n = 1287/9735), of which 50.7% cases were bilateral (Table 1). Pterygium was located nasally in 94.7% (n = 1837) of eyes, and double head pterygium was seen in 2.2% (n = 43) of the eyes. The extent of involvement was found to be grade 1 in 905 eyes (46.6%), grade 2 in 944 eyes (48.7%), and grade 3 in 59 eyes (3.1%). A rising trend of prevalence was observed with increasing age and the highest prevalence was observed in 60–69 years age group (15.8%) (Table 1). Males and females had similar prevalence (13.4% vs. 13.1%) (p = 0.636) overall and at across all age groups (Table 1). A significant difference was observed between the prevalence of pterygium at three study locations (p<0.001). Prakasam had the highest prevalence (20.3; CI 18.8–21.7) followed by Delhi NCR (11.2; CI 10.1–12.2) and Guwahati (9.1; CI 8.1–10.1).

On univariate analysis, pterygium was associated with older age, tropical and coastal location, lower levels of literacy, history of indoor smoke exposure, higher quintiles of lifetime cumulative effective sun exposure, use of headgear, and BMI <25kg/m$^2$ in the overall population (Table 2).

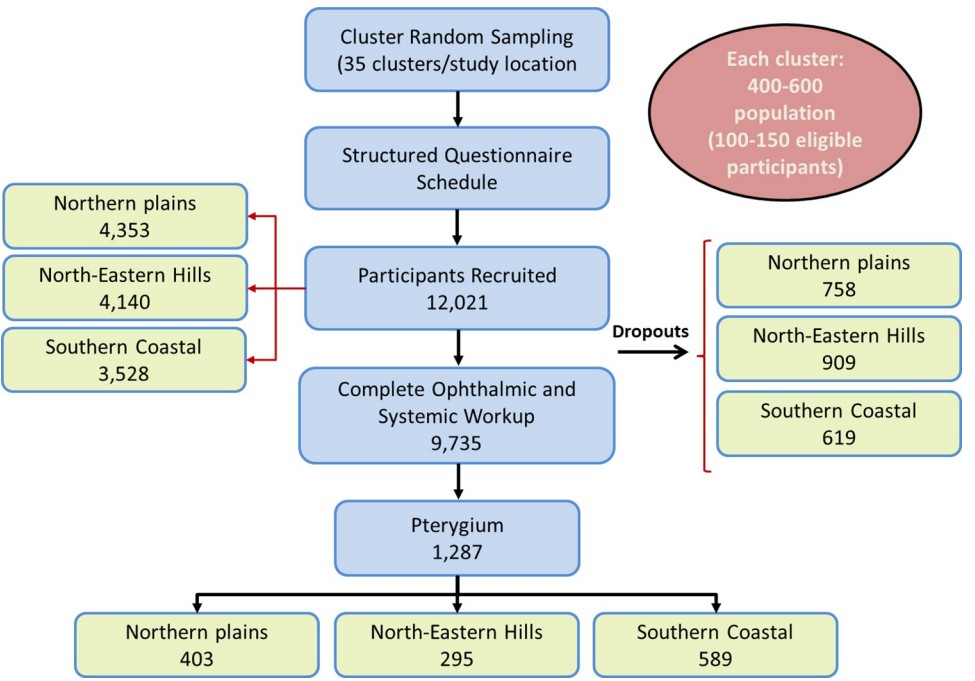

**Fig 1. Flowchart depicting participant enrolment and study process.**

**Table 1. Prevalence rates of pterygium in the study population by age and gender.**

| | | Overall | | | | Gender | | | |
| --- | --- | --- | --- | --- | --- | --- | --- | --- | --- |
| | | | | | | Males | | Females | |
| | N | Any pterygium % (CI) | Bilateral pterygium % (CI) | N | Any pterygium % (CI) | Bilateral pterygium % (CI) | N | Any pterygium % (CI) | Bilateral pterygium % (CI) |
| **All participants** | 9735 | 13.2 (12.5–13.9) | 6.7 (6.2–7.2) | 4426 | 13.4 (12.4–14.4) | 7.0 (6.2–7.7) | 5309 | 13.1 (12.1–14.0) | 6.5 (5.7–7.1) |
| **Age group** | | | | | | | | | |
| 40–49 years | 3998 | 11.1 (10.1–12.1) | 5.8 (5.0–6.5) | 1727 | 11.3 (9.8–12.8) | 6.4 (5.5–7.5) | 2271 | 11 (9.7–12.3) | 5.4 (4.4–6.3) |
| 50–59 years | 2438 | 13.7 (12.3–15.1) | 6.9 (5.8–7.8) | 1138 | 12.6 (10.6–14.5) | 6.2 (4.7–7.5) | 1300 | 14.8 (12.8–16.7) | 7.5 (6.0–8.8) |
| 60–69 years | 1981 | 15.8 (14.2–17.4) | 7.7 (6.5–8.8) | 900 | 17.1 (14.6–19.6) | 8.7 (6.8–10.5) | 1081 | 14.7 (12.6–16.8) | 6.9 (5.4–8.4) |
| 70+ years | 1318 | 14.7 (12.8–16.6) | 7.7 (6.2–9.1) | 661 | 15.3 (12.5–18.0) | 7.9 (5.8–9.9) | 657 | 14.2 (14.5–16.8) | 7.5 (5.4–9.4) |
| p-value | | *<0.001* | *0.015* | | *<0.001* | *0.075* | | *0.002* | *0.044* |

Occurrence of pterygium correlated with presence of hypermetropia, astigmatism, any cataract, especially nuclear cataract (P<0.0001) (Table 3). CUVAF imaging was performed in a subset of population of Delhi NCR (n = 1145) and Guwahati (n = 133) but no association with pterygium was observed (Table 3).

Multi-variable logistic regression analysis using backward stepwise elimination of variables was performed for all the factors showing significant association on univariate analysis in the overall population and at each study location (Table 4). In the overall population, pterygium was associated with study location, literacy levels, lifetime cumulative effective sun exposure and BMI. The study population at Prakasam (south coastal region) had the highest likelihood of pterygium (OR- 2.1; CI 1.8–2.5; p<0.001). Illiterates had about twice the risk (OR-1.7; CI 1–2.7; p = 0.037) of having pterygium than those educated up to and beyond graduation. Increasing lifetime cumulative sun exposure had a positive association with pterygia. In the overall population, the fifth quintile of lifetime cumulative effective sun exposure had about two times higher risk than first quintile (OR-2.3; CI 1.8–2.9; p<0.001). Variable associations were noted at different study locations. BMI $\geq$25kg/m$^2$ showed a protective effect for pterygium (OR-0.8; CI 0.7–0.9; p = 0.013). Indoor smoke exposure (OR-1.3; CI 1.1–1.7; p = 0.012), astigmatic refractive error (OR-1.4; CI 1.1–1.9; p = 0.017) and cortical cataract (OR-3.7; CI 1.4–9.5; p = 0.017) showed significant positive associations in population of Prakasam. DED was observed as a protective factor in Delhi NCR (OR- 0.7; CI 0.6–0.9; p = 0.006) and a risk factor in Prakasam (OR- 1.6; CI 1.2–2.2; p = 0.002).

## Discussion

The ICMR EYE SEE study is a multicentric, population-based study from India investigating the prevalence and associated risk factors of pterygium at distinct geographical locations. We found that coastal location, increasing lifetime cumulative effective sun exposure, illiteracy, and low BMI were significant risk factors for pterygium.

In our study, the prevalence of pterygium in any eye in adults aged $\geq$40 years in a rural population was 13.2%. The prevalence of pterygium was higher at the southern coastal site of Prakasam (20.3%). When compared with other studies from India, our result was higher than that reported by a study in South India (9.5%), Andhra Pradesh Eye Disease Study (APEDS) (11.7%), and a hospital-based series (10.5%) [17–19]. One study from central India has reported a pterygium rate of 15% in its rural population [20]. A similar rate of 15.2% was reported from a rural cohort in south India [17]. It is imperative to highlight that climatic conditions in India vary considerably with geographical location, and there are substantial

**Table 2. Association of pterygium with sociodemographic and systemic factors.**

| Variable | Overall OR (95%CI) | p-value | Delhi NCR OR (95%CI) | p-value | Guwahati OR (95%CI) | p-value | Prakasam OR (95%CI) | p-value |
|---|---|---|---|---|---|---|---|---|
| **Age** | | | | | | | | |
| 40–49 years | 1 | | 1 | | 1 | | 1 | |
| 50–59 years | 1.27 (1.1–1.48) | 0.002 | 1.28 (0.97–1.69) | 0.081 | 1.25 (0.92–1.69) | 0.147 | 1.2 (0.95–1.51) | 0.123 |
| 60–69 years | 1.49 (1.28–1.74) | <0.001 | 1.55 (1.17–2.04) | 0.002 | 1.34 (0.97–1.86) | 0.071 | 1.42 (1.12–1.8) | 0.004 |
| 70+ years | 1.38 (1.15–1.65) | 0.001 | 1.78 (1.32–2.4) | <0.001 | 1.28 (0.87–1.88) | 0.220 | 1.06 (0.79–1.42) | 0.679 |
| **Gender**, Male vs. Female | 0.97 (0.86–1.09) | 0.649 | 0.8 (0.65–0.98) | 0.033 | 0.65 (0.51–0.82) | <0.001 | 1.43 (1.19–1.72) | <0.001 |
| **Study location** | | | | | | | | |
| Delhi NCR | 1 | | - | - | - | - | - | - |
| Guwahati | 0.81 (0.68–9.31) | 0.005 | - | - | - | - | - | - |
| Prakasam | 2 (1.75–2.3) | <0.001 | - | - | - | - | - | - |
| **Education** | | | | | | | | |
| Graduation | 1 | | 1 | | 1 | | 1 | |
| High school (9–12) | 1.22 (0.74–2) | 0.431 | 0.87 (0.43–1.75) | 0.687 | 1.47 (0.62–3.49) | 0.383 | 1.89 (0.55–6.54) | 0.312 |
| Middle school (6–8) | 1.38 (0.82–2.31) | 0.220 | 0.97 (0.47–1.99) | 0.938 | 1.67 (0.67–4.15) | 0.272 | 2.16 (0.61–7.68) | 0.234 |
| Primary school (up to 5) | 1.76 (1.08–2.87) | 0.023 | 1.33 (0.66–2.68) | 0.431 | 1.42 (0.59–3.36) | 0.429 | 4.18 (1.28–13.65) | 0.018 |
| Illiterate | 2.56 (1.59–4.10) | <0.001 | 1.3 (0.67–2.53) | 0.441 | 1.79 (0.77–4.18) | 0.174 | 6.58 (2.06–21.07) | 0.001 |
| **Occupation**, Indoor vs. Outdoor | 0.92 (0.79–1.07) | 0.308 | 0.75 (0.57–0.97) | 0.029 | 1.98 (0.8–4.9) | 0.140 | 1.73 (1.42–2.12) | <0.001 |
| **Smoking**, No vs. Yes | 0.98 (0.87–1.11) | 0.787 | 1.28 (1.04–1.58) | 0.023 | 1.36 (1.04–1.79) | 0.025 | 0.71 (0.57–0.87) | 0.001 |
| **Indoor smoke exposure**, No vs. Yes | 0.86 (0.76–0.97) | 0.017 | 0.83 (0.67–1.02) | 0.086 | 0.99 (0.64–1.52) | 0.968 | 1.31 (1.09–1.58) | 0.004 |
| **Lifetime cumulative effective sun exposure** | | | | | | | | |
| 1st quintile | 1 | | 1 | | 1 | | 1 | |
| 2nd quintile | 1.65 (1.33–2.04) | <0.001 | 1.31 (0.89–1.91) | 0.172 | 1.38 (0.92–2.06) | 0.121 | 2.19 (1.56–3.1) | <0.001 |
| 3rd quintile | 1.83 (1.48–2.27) | <0.001 | 1.53 (1.06–2.23) | 0.023 | 1.35 (0.89–2.03) | 0.152 | 2.64 (1.87–3.72) | <0.001 |
| 4th quintile | 1.99 (1.62–2.46) | <0.001 | 1.78 (1.23–2.54) | 0.002 | 1.49 (0.85–1.94) | 0.228 | 2.97 (2.12–4.17) | <0.001 |
| 5th quintile | 2.62 (2.14–3.21) | <0.001 | 2.79 (1.98–3.93) | <0.001 | 1.77 (1.19–2.6) | 0.004 | 3.3 (2.36–4.59) | <0.001 |
| **Head gear** [*], No vs. Yes | 1.74 (1.13–2.68) | 0.011 | 1.16 (0.49–2.71) | 0.732 | 1.18 (0.71–1.96) | 0.534 | - | - |
| **Diabetes,** No vs. Yes | 1.01 (0.82–1.25) | 0.902 | 0.9 (0.57–1.43) | 0.656 | 1.14 (0.68–1.9) | 0.622 | 0.7 (0.54–0.92) | 0.011 |
| **Hypertension,** No vs. Yes | 0.9 (0.79–1.02) | 0.103 | 0.93 (0.75–1.15) | 0.501 | 1.01 (0.78–1.29) | 0.955 | 0.72 (0.59–0.87) | 0.001 |
| **BMI $\geq 25kg/m^2$**, No vs. Yes | 0.84 (0.73–0.97) | 0.017 | 0.85 (0.67–1.08) | 0.180 | 0.77 (0.53–1.12) | 0.167 | 0.61 (0.5–0.75) | <0.001 |

NCR, National capital region; BMI, Body mass index

[*]In Prakasam, all the participants with pterygium reported regular use of a headgear

differences in the amount of sunlight received, lifestyle preferences, and primary occupation as one moves from one geographical region to another.

The prevalence of bilateral pterygium in the current study was 6.7%. This was similar to that reported in Andhra Pradesh State, India (6.9%) and higher than that observed in Central India (4%) and Singapore (4.9%) [20, 21]. Literature review suggests considerable variations in pterygium rates across studies from various parts of the world. The prevalence in our study was higher than reported by studies in Greater Beijing, China (2%) [7], Victoria, Australia (2.8%) [5], and Singapore (12.3%) [23] and lower than the pterygium frequency in Indonesia (17%) [22], and rural Dali, China (29%) [8]. Racial and genetic differences along with behavioral and environmental variations between populations studied could explain this discordance [4, 6, 22].

Our study corroborates findings from other studies that show that the prevalence of pterygium increases with increasing age [17–21, 23]. The prevalence increased from 11.1% in 40–49

**Table 3. Association of pterygium with CUVAF and ophthalmological variables.**

| Variable | Overall OR (95%CI) | p-value | Delhi NCR OR (95%CI) | p-value | Guwahati OR (95%CI) | p-value | Prakasam OR (95%CI) | p-value |
|---|---|---|---|---|---|---|---|---|
| **Myopia**, No vs. Yes | 0.54 (0.47–0.61) | *<0.001* | 0.92 (0.71–1.21) | *0.569* | 0.74 (0.45–1.21) | *0.228* | 1.12 (0.87–1.45) | *0.368* |
| **Hypermetropia**, No vs. Yes | 1.87 (1.65–2.12) | *<0.001* | 1.08 (0.84–1.39) | *0.546* | 1.21 (0.79–1.86) | *0.374* | 1.21 (0.79–1.86) | *0.374* |
| **Astigmatism**, No vs. Yes | 1.23 (1.05–1.44) | *0.010* | 1.02 (0.8–1.3) | *0.880* | 1.24 (0.86–1.8) | *0.242* | 1.4 (1.07–1.82) | *0.014* |
| **Dry eye disease**, No vs. Yes | 0.97 (0.85–1.11) | *0.687* | 0.8 (0.64–0.99) | *0.038* | 1.49 (1.15–1.94) | *0.003* | 2.01 (1.54–2.62) | *<0.001* |
| **Any Cataract**, No vs. Yes | 1.32 (1.17–1.49) | *<0.001* | 1.32 (1.07–1.64) | *0.011* | 1.16 (0.89–1.51) | *0.272* | 1.12 (0.93–1.34) | *0.231* |
| **Cortical cataract**, No vs. Yes | 1.10 (0.87–1.39) | *0.422* | 1.42 (1.03–1.96) | *0.031* | 1.22 (0.79–1.86) | *0.370* | 4.71 (1.94–11.42) | *0.001* |
| **Nuclear cataract**, No vs. Yes | 1.32 (1.15–1.53) | *<0.001* | 1.48 (1.14–1.92) | *0.003* | 1.17 (0.87–1.57) | *0.309* | 1.11 (0.9–1.37) | *0.335* |
| **PSC**, No vs. Yes | 0.97 (0.72–1.3) | *0.818* | 1.14 (0.81–1.63) | *0.453* | 1.5 (0.67–3.35) | *0.324* | 0.84 (0.35–2.05) | *0.701* |
| **ARMD**, No vs. Yes | 0.89 (0.65–1.23) | *0.487* | 1.23 (0.86–1.77) | *0.263* | 0.81 (0.35–1.89) | *0.633* | 1.3 (0.14–12.55) | *0.819* |
| **CUVAF**[*] | | | | | | | | |
| 1st quintile | 1 | | 1 | | 1 | | - | |
| 2nd quintile | 1.1 (0.59–1.9) | *0.851* | 1.11 (0.6–2.06) | *0.734* | 0.51 (0.5–5.18) | *0.567* | - | |
| 3rd quintile | 1.25 (0.71–2.21) | *0.447* | 1.17 (0.63–2.17) | *0.622* | 1.89 (0.42–8.52) | *0.412* | - | |
| 4th quintile | 1.27 (0.72–2.25) | *0.406* | 1.25 (0.67–2.29) | *0.483* | 1.51 (0.31–7.26) | *0.610* | - | |

NCR, National capital region; OR, Odds Ratio; CI, Confidence Interval; PSC, Posterior subcapsular cataract; ARMD, Age related macular degeneration; CUVAF, Conjunctival ultraviolet autofluorescence

[*]CUVAF was not recorded in population of Prakasam

years age group to 15.8% in 60–69 years age group. The APEDS in India reported a similar trend wherein the prevalence was 11.4% in 40–49 years age group and increased to 15.6% in 60–69 years age group [18]. Similarly, Cajucom et al observed that the pterygium rate increased from 7.1% in 4th decade to 19% in 6th decade in Malay population of Singapore [21]. The combined effect of cumulative ocular UV damage and age-related changes in the ocular surface milieu predisposes older individuals to increased risk of pterygium.

In the current study, pterygium rates were similar between males and females (13.4% vs. 13.1%). While most studies report a higher prevalence in males, two studies from south India demonstrated no difference between the two sexes [3, 5, 7, 17, 18, 23]. Zhong et al and Lu et al even documented higher risk in females [8, 24]. Interestingly, in our study, site-specific analyses showed that males had higher pterygium rates at Delhi NCR (12.5% vs. 10.2%) and Guwahati (11.1% vs. 7.5%) but lower at Prakasam (17.2% vs. 22.8%). Men are traditionally more prone to occupational and recreational sun exposure, which can explain their higher risk but the varied results across studies suggest that other factors might be at play.

Exposure to sunlight, particularly UV radiation, is incriminated to be the most important risk factor for pterygium and all other factors are suspected to be proxy measures for it [16]. Despite its significance, there is no objective diagnostic tool for measurement of total amount of sun exposure of an individual. Most studies have used number of hours spent outdoors and outdoor occupation as a surrogate measure of sun exposure [18–20, 22, 25]. In the current study, we have used an individualized approach for calculating the approximate cumulative lifetime effective sunlight exposure taking into account the effect of protective headgear and eyewear with the help of Melbourne formula [16]. A stronger positive association was found between the higher cumulative effective sun exposure and pterygium. Our results support other studies in literature [5, 17, 26]. Asokan et al also utilized Melbourne model to calculate

**Table 4. Significant associations of pterygium on multivariate analysis.**

| Variable | Overall Adjusted OR (95% CI) | p-value | Delhi NCR Adjusted OR (95% CI) | p-value | Guwahati Adjusted OR (95% CI) | p-value | Prakasam Adjusted OR (95% CI) | p-value |
|---|---|---|---|---|---|---|---|---|
| **Study location** | | | | | | | | |
| Delhi NCR | 1 | | - | - | - | | - | |
| Guwahati | 1.01(0.84–1.2) | 0.954 | - | - | - | - | - | - |
| Prakasam | 2.11(1.83–2.45) | <0.001 | - | - | - | - | - | - |
| **Education level** | | | | | | | | |
| Graduation | 1 | | - | | - | | 1 | |
| High School (9–12) | 1.18(0.71–1.94) | 0.524 | - | - | - | - | 1.61(0.46–5.66) | 0.461 |
| Middle School (6–8) | 1.21(0.72–2.04) | 0.474 | - | - | - | - | 1.65(0.45–6.04) | 0.449 |
| Primary school (up to 5) | 1.37(0.84–2.25) | 0.212 | - | - | - | - | 2.92(0.88–9.69) | 0.080 |
| Illiterate | 1.67(1.03–2.71) | 0.037 | - | - | - | - | 3.85(1.18–12.6) | 0.026 |
| **Indoor smoke exposure**, No vs. Yes | - | | - | - | - | - | 1.33(1.07–1.66) | 0.012 |
| **Lifetime cumulative effective sun exposure** | | | | | | | | |
| 1st quintile | 1 | | 1 | | 1 | | 1 | |
| 2nd quintile | 1.45(1.15–1.82) | 0.001 | 0.98(0.61–1.57) | 0.931 | 1.45(1.06–2) | 0.421 | 1.85(1.16–2.96) | 0.010 |
| 3rd quintile | 1.52(1.22–1.89) | <0.001 | 1.32(0.86–2.03) | 0.198 | 1.23(0.85–1.79) | 0.264 | 2.1(1.44–3.06) | <0.001 |
| 4th quintile | 1.69(1.36–2.11) | <0.001 | 1.37(0.91–2.06) | 0.131 | 1.77(1.17–2.67) | 0.007 | 2.14(1.48–3.11) | <0.001 |
| 5th quintile | 2.28(1.82–2.85) | <0.001 | 2.36(1.62–3.45) | <0.001 | 2.15(1.16–4) | 0.015 | 2.4(1.62–3.56) | <0.001 |
| **BMI** | | | | | | | | |
| <25 kg/m2 | 1 | | - | | - | | 1 | |
| ≥25 kg/m2 | 0.82(0.71–0.95) | 0.009 | - | - | - | - | 0.8(0.64–1) | 0.052 |
| **Astigmatism**, No vs. Yes | - | - | - | - | - | - | 1.41(1.07–1.88) | 0.017 |
| **Dry eye disease**, No vs. Yes | - | - | 0.74(0.59–0.92) | 0.006 | - | - | 1.61(1.19–2.19) | 0.002 |
| **Cortical cataract**, No vs. Yes | - | - | - | - | - | - | 3.7(1.43–9.54) | 0.007 |

NCR, National capital region; BMI, Body mass index

lifetime sun exposure but only in a subset of participants [17]. Chun et al used serum 25(OH) D levels as an objective indicator of sun exposure and showed highest association with sun exposure of >5 hours/day [25].

Lower education level reflects lower socioeconomic status and serves as an indirect indicator of UV exposure as individuals with higher education are more likely to be involved in indoor skilled professions. Similar to previous literature, lower literacy levels were associated with pterygium in our study [20, 25]. Higher BMI had a protective effect on pterygium in our study. It could be suggested that people with higher BMI are more likely to be confined indoors with resultant lower sun exposure. Literature provides inconclusive evidence regarding the association between pterygium and BMI [27, 28]. McKnight et al reported that participants with pterygium were less likely to be over-weight/obese than those without it, although no association was found between overall BMI and pterygium [27]. On contrary, increased oxidative stress in obese has been impli-cated in pterygium occurrence in females [28]. The exact relationship between BMI and pterygium cannot be explained with the available evidence and prospective studies are required to establish the causality.

The prevalence of pterygium showed distinct variation in our study with respect to location. Highest prevalence was observed at Prakasam (Southern coast) (20.3%) followed by Delhi

NCR (Northern plains) (11.2%) and Guwahati (North-eastern hills) (9.1%). Climatic and environmental factors like sun exposure, air pollution and humidity, and lifestyle and socioeconomic differences may account for the observed difference. Prakasam being a coastal district sees a lot of fishing on the seas leading to higher exposure to UV radiation. It is well known that a negative association exists between latitude and pterygium prevalence [10]. This relationship has been explained partly by the diminishing UV component of solar radiation with increasing latitude [10]. Although Wong et al refuted this theory by demonstrating similar pterygium rates in two populations at notably different latitudes, we found higher prevalence of pterygium at Prakasam, that was located closest to equator (15˚ N) when compared to Delhi NCR (28.7˚ N) and Guwahati (26.1˚ N) [4]. We also noted that even though the median lifetime sun exposure was highest in Delhi NCR, pterygium rate was highest in Prakasam. As per our previously published results, Prakasam was the site with highest mean UVA and UVB exposure, maximum average wind speed, and highest humidity, and lowest air pollution [12, 13] Lee et al found no association between air pollution and pterygium [29]. These observations highlight that a complex interplay among environmental parameters underlies the pathogenesis of pterygium and air quality parameters could play a role, although individualized data will give more valuable insight.

On site-specific analysis, we observed that indoor smoke exposure, astigmatic refractive error, DED and cortical cataract were associated with higher odds of pterygium in population of Prakasam. We have previously reported that cortical cataract is strongly associated with UV exposure [12]. Indoor smoke exposure due to wood and biomass fuel for cooking and heating alters ocular surface health [30]. Its role in causation of DED is well established [13]. DED is an established risk factor for pterygium and unevenness of ocular surface due to pterygium predisposes to DED [23, 29]. Counterintuitive results observed with DED in Delhi NCR only substantiate the theory of interplay of multiple factors that may be responsible for the etiopathogenesis of pterygium and that no definite relationship has yet been established.

CUVAF imaging has recently been developed to serve as an objective biomarker of ocular UV exposure. Studies from Australia have reported increasing CUVAF as an independent risk factor for pterygium [31, 32]. Taking a holistic approach in evaluating risk factors for pterygium, we performed CUVAF imaging in a subset of participants but found no association between the two. We feel that capturing CUVAF images in a large population-based study like the present one is cumbersome and not practically feasible.

The strengths of our study are the large multicentric population-based sample size, high response rate, individualized approach to measure lifetime cumulative sun exposure, and comprehensive assessment of risk factors. Limitations include the inability to determine individualized air quality parameters and the cross-sectional design of the study.

In conclusion, this study reports a high prevalence of pterygium in rural populations of India. Higher prevalence of pterygium was associated with coastal location, increasing lifetime cumulative effective sun exposure, lower literacy levels, and lower BMI in this study. To the best of our knowledge, this is the largest population-based study that highlights ocular morbidity due to pterygium and its associated risk factors at diverse geographical locations. This study provides further evidence and support to the theory that pterygium is a multifactorial ocular disorder caused by complex interactions between multitudes of intrinsic and extrinsic factors. Modifiable risk factors should be targeted to reduce the morbidity associated with this condition so that the high burden, especially in tropical and subtropical regions, may be tackled effectively.

## Acknowledgments

Dr. Saurabh Agarwal Jwalaprasad, Dr. Bhagbat Nayak, Dr. Jayanta Thakuria, Dr. Indrani Goswami, Ms. Tanya Patel, Ms. Ankita Mall, Dr. Rupesh M Das are acknowledged for their contribution to data acquisition. Mr. Amit Bhardwaj and Mr. Deepak Kumar are acknowledged for their contribution to data management and analysis. We would like to acknowledge the ICMR Task Force on global climate change and health chaired by Prof. Seyed E. Hasnain, IIT Delhi, for periodic review and technical inputs during the course of the study. All the members of the ICMR Eye Sun Exposure & Environment "EYE SEE" study group are acknowledged for their contributions to the project.

THE ICMR EYE SUN EXPOSURE & ENVIRONMENT "EYE SEE" STUDY GROUP.

| Centres | Principal investigator | Co-investigators | Scientist/Research officers |
|---|---|---|---|
| Dr. Rajendra Prasad Centre for Ophthalmic Sciences, AIIMS, New Delhi (Coordinating Center) | Dr. Radhika Tandon Dr. Praveen Vashist | Dr. Noopur Gupta Dr. Vivek Gupta | Dr. Saumya Yadav Dr.Pranita Sahay Dr.Rashmi Singh Dr. Meenakshi Wadhwani Dr. Shweta Dr. Aparna Gupta Dr. Saurabh Agarwal Jwalaprasad Dr. Bhagbat Nayak |
| Indian Institute of Public Health, Hyderabad | Dr. GVS Murthy | Dr. K. Vishwanath | Dr. Hemant Kumar Dr. Vijay Kiran |
| Regional Institute of Ophthalmology, Guwahati | Dr. C.K.Barua Dr. Dipali Deka | | Dr. Jayanta Thakuria Dr. Indrani Goswami |
| National Physical Laboratory, New Delhi | Dr. Sachchidanand Singh | | Ms. Tanya Patel Ms. Ankita Mall Dr. Rupesh M Das |

## Author Contributions

**Conceptualization:** Radhika Tandon.

**Data curation:** Radhika Tandon, Praveen Vashist, Noopur Gupta, Vivek Gupta, Saumya Yadav, Dipali Deka, Sachchidanand Singh, K. Vishwanath, G. V. S. Murthy.

**Formal analysis:** Radhika Tandon, Praveen Vashist, Vivek Gupta, Saumya Yadav, Sachchidanand Singh, K. Vishwanath, G. V. S. Murthy.

**Funding acquisition:** Radhika Tandon.

**Investigation:** Radhika Tandon, Praveen Vashist, Noopur Gupta, Sachchidanand Singh, K. Vishwanath, G. V. S. Murthy.

**Methodology:** Radhika Tandon, Praveen Vashist, Noopur Gupta, Sachchidanand Singh, G. V. S. Murthy.

**Project administration:** Radhika Tandon, Praveen Vashist, Dipali Deka, Sachchidanand Singh, G. V. S. Murthy.

**Supervision:** Radhika Tandon, Praveen Vashist, Dipali Deka, Sachchidanand Singh, G. V. S. Murthy.

**Writing – original draft:** Saumya Yadav.

**Writing – review & editing:** Radhika Tandon, Praveen Vashist, Noopur Gupta, Vivek Gupta, Dipali Deka, Sachchidanand Singh, K. Vishwanath, G. V. S. Murthy.

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
