## [Decision Letter · Decision Letter 0]

15 Mar 2022

PONE-D-21-32060Study of the association of sun exposure, ultraviolet radiation effects and other risk factors for pterygium (The SURE RISK for Pterygium Study) in geographically diverse adult (≥ 40 years) rural populations of India 3rd report of ICMR EYE SEE groupPLOS ONE

Dear Prof. Tandon

Thank you for submitting your manuscript to PLOS ONE. After careful consideration, we feel that it has merit but does not fully meet PLOS ONE’s publication criteria as it currently stands. Therefore, we invite you to submit a revised version of the manuscript that addresses the points raised during the review process.

We look forward to receiving your revised manuscript.

Kind regards,

Muralidhar M. Kulkarni

Academic Editor

PLOS ONE

Journal Requirements:

3. Thank you for stating the following in the Financial support Section of your manuscript:

“Indian Council of Medical Research (ICMR), India provided funding for this research project (Grant No 68/4/2009-NCD-1). The funding source had no role in the study design, in the collection, analysis and interpretation of data, writing of the report; and in the decision to submit the article for publication.”

We note that you have provided additional information within the Financial support  Section that is not currently declared in your Funding Statement. Please note that funding information should not appear in other areas of your manuscript. We will only publish funding information present in the Funding Statement section of the online submission form.

“RT (Grant No 68/4/2009-NCD-1) Indian Council of Medical Research”

“RT (Grant No 68/4/2009-NCD-1) Indian Council of Medical Research”

Additional Editor Comments (if provided):

I congratulate the authors for this important work. It will add to the existing knowledge and herald further research.

Minor corrections required.

Authors can address the reviewers concerns and consider the following.

The methodology can be condensed by using appropriate references so that it is crisp and less time consuming for readers to go to the crux of the study.

Reviewers' comments:

Reviewer's Responses to Questions

**Comments to the Author**

1. Is the manuscript technically sound, and do the data support the conclusions?

Reviewer #1: Yes

2. Has the statistical analysis been performed appropriately and rigorously? 

Reviewer #1: Yes

3. Have the authors made all data underlying the findings in their manuscript fully available?

Reviewer #1: Yes

4. Is the manuscript presented in an intelligible fashion and written in standard English?

Reviewer #1: Yes

5. Review Comments to the Author

Reviewer #1: This study has generated few highlighting information regarding pterygium and its association with several risk factors. It definitely adds more knowledge about the disease. I have few questions to the author.

1) The lifetime cumulative effect of sun exposure was calculated with the help of satellite-based data for environmental UV exposure. How relevant and accurate is the data? Is the data for entire lifetime was calculated based on the current location where the study took place? Also AOD values and meteorological data was used in the study from three mentioned locations. Are those accounted at the time of examination? Or any time period? How the author will justify it based on the participants occupation location? As some might have different work place where they might be exposed more than that particular location where the measurement was done. It will be better if explained elaborately.

2) Is there any association between pterygium and the type of work the participants does?

3) Why indoor smoke exposure was considered? Does it include passive smokers as well?

6. PLOS authors have the option to publish the peer review history of their article (what does this mean?). If published, this will include your full peer review and any attached files.

Reviewer #1: **Yes: **Avinash Soundararajan

---

## [Author Response · Author response to Decision Letter 0]

26 Apr 2022

Reviewer’s Comments 

Q1. The lifetime cumulative effect of sun exposure was calculated with the help of satellite-based data for environmental UV exposure. How relevant and accurate is the data? Is the data for entire lifetime was calculated based on the current location where the study took place? Also AOD values and meteorological data was used in the study from three mentioned locations. Are those accounted at the time of examination? Or any time period? How the author will justify it based on the participants occupation location? As some might have different workplace where they might be exposed more than that particular location where the measurement was done. It will be better if explained elaborately. 

A1. We thank the reviewer for bringing up this important point. The estimation of lifetime effective sun exposure was not satellite-based but was calculated for every participant in the study using the following formula, based on the Melbourne visual impairment project model: 

Lifetime Effective Sun Exposure = Σ [Daily hours of sun exposure without headgear + (Daily hours of sun exposure using headgear x protection factor)] x 365 x Number of years. 

The study location has no role in the determination of lifetime cumulative effective sun exposure. We can assure the reviewer that this data is reliable and the Melbourne model has been successfully used by several studies for calculating lifetime ocular sun exposure. 

Satellite-based data for the entire region was used for the measurements of AOD, UVA and UVB flux at the three study locations for the study period i.e. between 2010 and 2016. In addition, meteorological data for humidity, precipitation, temperature, wind speed, and air pollutants were also obtained for the three locations for the study period. 

We agree with the reviewer that the topics of lifetime cumulative exposure and climatic parameters need an elaborate explanation. These topics have been discussed in detail in our previous publications (mentioned below). To keep the manuscript concise and easy to read we decided not to discuss it in depth in the current manuscript and have provided appropriate references for the same. 

1. Vashist P, Tandon R, Murthy GVS, Barua CK, Deka D, Singh S, Gupta V, Gupta N, Wadhwani M, Singh R, Vishwanath K; ICMR-EYE SEE Study Group. Association of cataract and sun exposure in geographically diverse populations of India: The CASE study. First Report of the ICMR-EYE SEE Study Group. PLoS One. 2020 Jan 23;15(1):e0227868. doi: 10.1371/journal.pone.0227868. 

2. Tandon R, Vashist P, Gupta N, Gupta V, Sahay P, Deka D, Singh S, Vishwanath K, Murthy GVS. Association of dry eye disease and sun exposure in geographically diverse adult (≥40 years) populations of India: The SEED (sun exposure, environment and dry eye disease) study - Second report of the ICMR-EYE SEE study group. Ocul Surf. 2020 Oct;18(4):718-730. doi: 10.1016/j.jtos.2020.07.016. 

Q2. Is there any association between pterygium and the type of work the participants does?

A2. We thank the reviewer for this comment. In this study, the occupation of the participants was classified primarily as indoors and outdoors. This is pertinent that there might be some variation in the specific job that an individual does but our predominant focus was to gather information on whether the individual had sunlight exposure during his working hours. What mattered was overall time spent outdoors and not only related to occupation. In the overall population, we did not find any association between pterygium and whether the occupation of participants involved indoor or outdoor activity. Page no: 9, Table 2. 

Q3. Why indoor smoke exposure was considered? Does it include passive smokers as well?

A3. We thank the reviewer for highlighting this important point. The indoor smoke exposure due to the use of biomass fuel is a major contributor to indoor (household) pollution and is associated with a number of acquired systemic diseases. For this study, indoor smoke exposure was defined as a lifetime history of use of biomass fuels (coal, dung-cakes, wood) in the kitchen. A positive association of indoor smoke exposure with cataract and dry eye disease has been found in our previous analyses. Our aim was to find any such possible association between pterygium and indoor smoke exposure. 

In order to account for the similar nature of exposure and mechanism of action on the eyes, passive smokers were included in the group “smokers” along with active smokers and not with the participants exposed to indoor smoke.

---

## [Decision Letter · Decision Letter 1]

3 Jun 2022

The association of sun exposure, ultraviolet radiation effects and other risk factors for pterygium (The SURE RISK for Pterygium Study) in geographically diverse adult (≥40 years) rural populations of India -3rd report of the ICMR-EYE SEE study group

PONE-D-21-32060R1

Dear Dr. Tandon, 

We’re pleased to inform you that your manuscript has been judged scientifically suitable for publication and will be formally accepted for publication once it meets all outstanding technical requirements.

Kind regards,

Muralidhar M. Kulkarni

Academic Editor

PLOS ONE

Additional Editor Comments (optional):

We are glad to inform you that the manuscript is accepted for publication.

Reviewers' comments:

Reviewer's Responses to Questions

**Comments to the Author**

1. If the authors have adequately addressed your comments raised in a previous round of review and you feel that this manuscript is now acceptable for publication, you may indicate that here to bypass the “Comments to the Author” section, enter your conflict of interest statement in the “Confidential to Editor” section, and submit your "Accept" recommendation.

Reviewer #1: All comments have been addressed

2. Is the manuscript technically sound, and do the data support the conclusions?

Reviewer #1: Yes

3. Has the statistical analysis been performed appropriately and rigorously? 

Reviewer #1: Yes

4. Have the authors made all data underlying the findings in their manuscript fully available?

Reviewer #1: Yes

5. Is the manuscript presented in an intelligible fashion and written in standard English?

Reviewer #1: Yes

6. Review Comments to the Author

Reviewer #1: The authors have addressed all my questions with appropriate response. This paper will definitely add up to the existing scientific knowledge.

7. PLOS authors have the option to publish the peer review history of their article (what does this mean?). If published, this will include your full peer review and any attached files.

Reviewer #1: **Yes: **Avinash Soundararajan

---

## [Editor Report · Acceptance letter]

12 Jul 2022

PONE-D-21-32060R1 

The association of sun exposure, ultraviolet radiation effects and other risk factors for pterygium (The SURE RISK for Pterygium Study) in geographically diverse adult (≥40 years) rural populations of India -3rd report of the ICMR-EYE SEE study group 

Dear Dr. Tandon:

I'm pleased to inform you that your manuscript has been deemed suitable for publication in PLOS ONE. Congratulations! Your manuscript is now with our production department. 

Kind regards, 

on behalf of

Dr. Muralidhar M. Kulkarni 

Academic Editor

PLOS ONE